# Gastrointestinal Journey of Human Milk Oligosaccharides: From Breastfeeding Origins to Functional Roles in Adults

**DOI:** 10.3390/microorganisms14010029

**Published:** 2025-12-22

**Authors:** Yosuke Komatsu, Megumi Furuichi, Takeshi Kokubo

**Affiliations:** Institute of Health Science, Kirin Holdings Co., Ltd., 2-26-1 Muraoka-Higashi, Fujisawa 251-8555, Japan

**Keywords:** human milk oligosaccharides, 2′-fucosyllactose, 3′-sialyllactose, 6′-sialyllactose, gut microbiota, prebiotics, intestinal barrier, short-chain fatty acids, adult nutrition

## Abstract

Human milk oligosaccharides (HMOs) are the third most abundant solid component in human milk and play crucial roles in shaping the gut microbiome and promoting infant health. Although their functions during infancy are well established, emerging evidence suggests that HMOs exert region-specific effects throughout the gastrointestinal tract, extending their benefits beyond early life. This review summarizes current findings on HMO activity in the oral cavity, stomach, small intestine, and large intestine, focusing on their microbiota-modulating, barrier-enhancing, and immunoregulatory effects. In the oral cavity, HMOs inhibit pathogen adhesion and biofilm formation, maintaining oral homeostasis. In the stomach, fucosylated and sialylated HMOs act as soluble decoy receptors, preventing Helicobacter pylori infection. In the small intestine, HMOs strengthen epithelial integrity, regulate inflammation, and promote nutrient absorption. In the large intestine, they serve as selective prebiotics for beneficial microbes, enhancing short-chain fatty acid production and improving barrier function. Although preclinical and clinical studies demonstrate their safety and efficacy, further research is required to elucidate their mechanisms in adults. Overall, HMOs represent multifunctional bioactive glycans with promising applications for gastrointestinal health across all ages.

## 1. Introduction

Human milk oligosaccharides (HMOs) are a structurally diverse group of free glycans present at high concentrations in human milk, ranking as the third most abundant solid component after lactose and lipids. Over 200 distinct HMOs have been identified, with concentrations reaching 20–25 g/L in colostrum and 10–15 g/L in mature milk [1,2,3]. HMOs exhibit structural diversity based on 5 monosaccharide units: glucose, galactose, N-acetylglucosamine, fucose, and sialic acid. They are primarily classified into 3 categories: neutral fucosylated HMOs (e.g., 2′-fucosyllactose [2′-FL] and 3-FL), neutral non-fucosylated HMOs (e.g., lacto-N-tetraose [LNT] and lacto-N-neotetraose [LNnT]), and acidic HMOs (e.g., 3′-sialyllactose [3′-SL] and 6′-SL), as shown in Figure 1 [4,5,6]. HMOs resist host digestion and reach the colon largely intact, where they perform critical functions in shaping the infant gut microbiome. Their most established role is as prebiotics, selectively promoting the growth of beneficial bacteria, such as *Bifidobacterium longum* subsp. *infantis* [7,8,9]. HMOs also inhibit microbial adhesion to the gut, modulate immune responses, and induce differentiation of intestinal epithelial cells, offering protection against infection by various pathogens [10,11,12,13,14,15]. Since conventional infant formula, which uses cow milk as a primary ingredient, does not naturally contain HMOs, formulas supplemented with HMOs have been developed to offer similar physiological benefits. HMOs in infant formula promote health outcomes akin to those of human milk. Formulas containing 2′-FL and LNnT, for example, promote bifidobacterial growth and alter gut microbiota to resemble that of breastfed infants [16]. Additionally, the intake of infant formula supplemented with 2′-FL has been proven to reduce gastrointestinal and respiratory infections and lower antibiotic use [17]. Infants consuming 2′-FL-enriched formula also exhibit lower plasma TNF-α levels and immune modulation comparable to breastfed infants [18]. Multiple safety trials have confirmed that growth parameters, including weight, length, and head circumference, are comparable to those of breastfed infants, with no adverse effects [19]. Furthermore, stool characteristics and gastrointestinal symptoms showed no significant differences, indicating no adverse impact on digestive function [17]. These findings highlight that HMO-supplemented formula offers benefits similar to human milk, including microbiome modulation, infection prevention, and immune regulation, all while ensuring safety. Additionally, HMOs are believed to support infant brain development through their effects on gut microbiota and synaptic plasticity. Berger et al. integrated 6 observational studies and found that exposure to fucosylated and sialylated HMOs was positively associated with cognitive, language, and motor development in infants aged 18–24 months [20]. Specifically, 2′-FL was significantly associated with higher cognitive scores at 24 months, and 6′-SL correlated with improved motor and cognitive outcomes. Fan et al. reviewed 26 preclinical and human studies and demonstrated that fucosylated and sialylated HMOs contribute to learning, memory, executive function, and white matter maturation via microbiota-mediated neurotrophic mechanisms [21]. Together, these findings suggest that HMOs are promising nutritional modulators for supporting optimal neurodevelopment during early life. The growing interest in HMO applications is driven by advances in enzymatic synthesis and microbial fermentation, which enable the large-scale production of selected HMOs. Although more than 200 distinct HMO structures have been identified in human milk, only a limited number are used in commercial products, with 2′-FL, 3′-SL, and 6′-SL being the most widely studied and applied [22,23,24]. The regulatory framework governing the use of these HMOs as food ingredients, including safety assessment and conditions of use, has been comprehensively reviewed by Salminen [25]. Regulatory authorities, including the European Food Safety Authority (EFSA) and the U.S. Food and Drug Administration (FDA), have approved several HMOs as novel food ingredients and substances generally recognized as safe (GRAS), authorizing their use in functional foods and dietary supplements [26,27,28].

Growing interest in the physiological roles of HMOs has accelerated research into their dynamic behavior and region-specific effects along the gastrointestinal tract [29]. Recent evidence from in vitro models, animal studies, and clinical trials suggests that HMOs influence host–microbe interactions throughout the gastrointestinal tract, from the oral cavity to the large intestine, potentially exerting functions in each region. While evidence in HMO functionality in infants is well established, clinical data in adults remain limited. Regulation of the gut microbiota is also critical in adults, and recent findings indicate that HMOs may support gut health and reduce inflammation [30]. However, systematic investigations into these effects in adults are still lacking, particularly those addressing region-specific effects across the entire gastrointestinal tract [31,32,33]. The aim of this review is to bridge this gap by synthesizing current knowledge on HMOs along the gastrointestinal tract, with a focus on region-specific effects. We integrated findings from in vitro, in vivo, and clinical studies across the oral cavity, stomach, small intestine, and large intestine, considering both microbial and host responses. Furthermore, we highlight emerging insights extending HMO applications beyond infancy to adults, underscoring their potential as functional ingredients across age groups.

## 2. Oral Section

### Contribution of HMOs to Oral Health

The oral cavity is the first site of interaction between HMOs and the host environment, yet it has historically received less attention than the lower digestive tract. Recent studies indicate that HMOs help regulate the oral microbiota and inhibit pathogen colonization. Several in vitro studies have shown that HMOs have anticariogenic properties. For example, *Streptococcus mutans*, a primary etiological agent of dental caries, showed reduced adhesion to glass surfaces via exopolysaccharides in the presence of 2′-FL [34]. Similarly, experiments using neonatal saliva demonstrated that 3′-SL significantly decreased biofilm formation by *S. mutans* [35]. Brassart et al. [36] further reported that, in an in vitro system, the H-type glycan structure (α1-2 linkage) in HMOs inhibited the adhesion of *Candida albicans* to human oral epithelial cells by approximately 55%. This inhibitory effect was specific to milk-derived glycopeptides and oligosaccharides, as Lewis-type fucosylated structures (α1-3 linkage or α1-4 linkage) did not exhibit similar activity [36]. More recently, Faria et al. demonstrated that 3′-SL inhibited *C. albicans* biofilm formation in vitro, supporting its potential as an antifungal component [37]. These effects are presumed to result from HMOs acting as soluble receptor analogs that competitively inhibit microbial adhesion to host glycan receptors. In a murine model, oral administration of 2′-FL reduced colonization by oral pathogens and attenuated cytokine induction in the intestinal mucosa, indicating reduced inflammation [38]. Collectively, these findings suggest that HMOs may help maintain oral homeostasis by inhibiting pathogen adhesion and modulating inflammatory responses. However, clinical trials directly evaluating the impact of HMOs on oral health are lacking. By comparison, randomized controlled trials using probiotics have shown reductions in *C. albicans* colonization, suggesting that HMOs may confer similar benefits [39,40]. Recent reviews have highlighted the potential of HMOs to exert anti-adhesive and antimicrobial effects in the oral environment; however, current evidence is limited to in vitro and animal studies, and human clinical data remain absent [29]. Thus, at present, conclusions regarding the effects of HMOs on oral health are largely mechanistic and based on findings from in vitro and other experimental model studies, and their actual clinical impact on oral health remains hypothetical. Large-scale clinical trials with extended follow-up periods are required to determine whether these mechanistic effects translate into measurable improvements in oral health.

## 3. Stomach Section

### Preventive Effects of HMOs on Helicobacter pylori Infection

After completing their function in the oral cavity, HMOs pass through the esophagus and reach the stomach. Although largely resistant to digestion in the upper gastrointestinal tract and primarily active in lower gut, recent studies have suggested that they may also exert biological effects in the stomach, particularly through interactions with pathogens, such as *H. pylori*. This bacterium colonizes the gastric mucosa and is associated with chronic gastritis, peptic ulcer disease, and gastric cancer [41,42]. Several in vitro studies have demonstrated that specific HMOs inhibit *H. pylori* adhesion to gastric epithelial cells. For example, 3′-SL interacts with the outer membrane protein HPAA, strongly blocking bacterial attachment [43]. Simon et al. further showed that sialylated oligosaccharides, particularly 3′-SL, significantly inhibited *H. pylori* adhesion to human gastric epithelial cells. 3′-SL not only prevented attachment of fresh clinical isolates but also detached previously adhered bacteria [44]. An in vivo study using a primate model demonstrated that administration of 3′-SL reduced *H. pylori* colonization in the stomach, supporting its potential as a preventive agent [45]. *H. pylori* preferentially recognizes α2-3-linked sialic acid, suggesting that HMOs may act as decoy receptors to competitively inhibit bacterial adhesion [46,47]. The fucosyltransferase gene *Fut2* regulates gastric mucin glycosylation, which is crucial for *H. pylori* adhesion. In FUT2-deficient mice, reduced α1,2-fucosylation of mucin was associated with decreased bacterial adhesion [48]. Similarly, Xu et al. introduced human α1-3/4-fucosyltransferase into goat mammary glands to produce “humanized” milk containing fucosylated antigens. This milk inhibited the binding of *H. pylori* antigens by up to 83%, suggesting that fucosylated glycans may contribute to infection prevention [49]. In a double-blind, randomized, placebo-controlled clinical trial involving 65 *H. pylori*-infected patients, 3’-SL administration (10 or 20 g/day for 4 weeks) significantly reduced bacterial load without serious adverse events, though complete eradication was not achieved. These findings indicate that while 3′-SL is safe, its efficacy as monotherapy is limited [50]. The limited clinical efficacy of HMOs observed so far against *H. pylori* infection is probably influenced by several factors. First, after oral administration, HMOs may not reach or remain at sufficiently high concentrations on the gastric mucosal surface, which could limit their ability to inhibit *H. pylori* adhesion through competitive binding to host receptors. Second, strain-to-strain differences in *H. pylori* adhesins and virulence factors, as well as in commensal microbes, are likely to result in variable sensitivity to specific HMOs. Third, when HMOs are administered as adjuncts to standard eradication regimens containing proton pump inhibitors and antibiotics, profound changes in the gastric environment and microbiota may mask any additional HMO-mediated benefits. Therefore, future clinical trials should be designed to rigorously define dose–response relationships for individual HMOs, to characterize strain-specific effects, and to account for inter-individual variability in the human microbiome when interpreting treatment outcomes.

## 4. Small Intestine Section

### 4.1. Barrier-Enhancing Effects of HMOs in the Small Intestine

HMOs that pass through the stomach intact reach the small intestine. Holscher et al. demonstrated that 2′-FL, 3-FL, and 6′-SL promote the differentiation of small intestinal epithelial cells [51]. Kong et al. reported that 3-FL increases hyaluronic acid and heparan sulfate, key components of the glycocalyx layer in Caco-2 cells, thereby enhancing barrier function [52]. Similarly, Cheng et al. found that 3-FL and LNT promote *Muc2* gene expression and protein secretion in goblet cells exposed to IL-13 and TNF-α, strengthening mucus layer integrity under inflammatory conditions [53]. These effects collectively reduce intestinal permeability and suppress inflammation. Wu et al. further showed that HMOs promote *Muc2* expression and restore goblet cell numbers, reconstructing the mucus layer and lowering intestinal permeability [54,55]. HMOs also induce endoplasmic reticulum chaperones such as protein disulfide isomerase in intestinal epithelial cells, ensuring proper folding and secretion of MUC2 and maintaining barrier integrity [56]. Enhanced barrier function and reduced inflammation contribute to protection against necrotizing enterocolitis (NEC). In mouse and piglet models, Sodhi et al. showed that 2′-FL and 6′-SL inhibit TLR4 activation and reduce expression of pro-inflammatory cytokines such as IL-6 and IL-8, thereby preventing NEC onset [57]. Despite strong preclinical evidence, few human studies have directly evaluated small intestinal barrier function. Common assessment methods include the lactulose/mannitol ratio and blood biomarkers, among which zonulin is a key regulator of tight junction dynamics and a sensitive indicator of intestinal permeability [58,59]. However, validated, non-invasive biomarkers for monitoring gut barrier integrity in clinical settings remain limited. Clinical trials that incorporate these parameters as predefined endpoints, and that further develop and validate additional biomarkers, are required to clarify the barrier-enhancing effects of HMOs in humans.

### 4.2. Immunomodulatory Effects of HMOs

HMOs enhance immune functions in the small intestine. He et al. demonstrated that 2′-FL suppresses the invasion of enterotoxigenic *Escherichia coli* and reduces IL-8 secretion in vitro experiments using the human ileum-derived epithelial cell line HCT8 and the immature small intestinal model H4 cells [60]. Cheng et al. showed that LNT activates multiple TLRs expressed on immune cells in the small intestine and promotes the secretion of IL-10 and TNF-α, indicating that HMOs may regulate immune responses in a structure-dependent manner [61]. Using 3D organoids derived from mouse ileal crypts, Wang et al. [62] found that lipopolysaccharide stimulation increased TLR4 expression and suppressed cell proliferation, whereas HMO treatment significantly reduced TLR4 levels and restored Ki67-positive proliferating cells. These findings suggest that HMOs attenuate intestinal inflammation and promote epithelial regeneration [62]. Li et al. further elucidated molecular mechanisms underlying HMO-induced immune enhancement in small intestinal epithelial cells. In in vitro and in vivo analyses, pooled HMO administration to Caco-2 cells suppressed proliferation-related genes (*Cdc6*, *Cdc20*) and upregulated differentiation markers (*Muc13*, *Apoa4*, *Hmgcs2*). Activation of the mTOR and PPAR pathways improved epithelial differentiation and barrier integrity. Correspondingly, HMOs increased MUC2-positive cells in the ileum, promoting intestinal regeneration in NEC model mice [63]. The immune-enhancing properties of HMOs may also help prevent severe infections. In a mouse influenza vaccination model, Xiao et al. [64] demonstrated that co-administration of 2′-FL and prebiotics (scGOS/lcFOS) enhanced vaccine-specific serum IgG levels and increased Th1 and regulatory T cell populations. These immune changes correlated with intestinal short-chain fatty acid (SCFA) concentrations and microbiota composition [64]. HMOs may additionally protect against food allergies. In an ovalbumin-sensitized mouse model, Castillo-Courtade et al. reported that oral administration of 2′-FL and 6′-SL alleviated allergic symptoms and increased the number of regulatory T cells in Peyer’s patches. 6′-SL also enhanced IL-10 production and suppressed *Tnf* expression in splenocytes, indicating that HMOs may mitigate allergic responses through regulatory T cells induction and mast cell stabilization [65]. Collectively, these in vitro and in vivo findings suggest that HMOs enhance immune function in the small intestine and contribute to the modulation of inflammation, defense against infection, and the attenuation of allergic responses. Evidence from human studies, however, remains limited. In a randomized controlled trial, Goehring et al. reported that infants receiving 2′-FL-supplemented formula exhibited lower levels of pro-inflammatory cytokines (IL-1, IL-6, TNF-α) and immune profiles resembling those of breastfed infants, suggesting a reduction in systemic inflammation [18]. A review of multiple randomized controlled trials by Reverri et al. further confirmed the safety and tolerability of 2′-FL-supplemented formulas and identified potential immune-related benefits, including reduced infection risk and improvements in inflammatory markers [66]. Current clinical studies primarily assess systemic immune indicators such as serum cytokines, immunoglobulins, and infection outcomes. However, direct evaluation of local small intestinal immune markers, including mucosal IgA, Peyer’s patch responses, and epithelial TLR signaling, remains limited owing to technical and ethical constraints. Establishment of reliable, non-invasive biomarkers will be essential for real-time monitoring of intestinal immune status in humans.

### 4.3. Effects of HMOs on Improving Nutrient Absorption

Recent evidence suggests that 2′-FL may directly affect the structure and function of the small intestine, independent of microbiome-mediated mechanisms. Paone et al. investigated the effects of 2′-FL on intestinal function in high-fat diet–induced obese C57BL/6 mice using transcriptomic analysis. They reported upregulation of *Lyz1*, *Reg3g*, and *Proglucagon* in small intestinal tissue, indicating enhanced epithelial activity. These transcriptional changes suggest that 2′-FL stimulates digestive and absorptive functions, thereby promoting nutrient uptake and metabolic efficiency through activation of intestinal epithelial metabolism [67]. In a human intestinal epithelial model, Holscher et al. found that 2′-FL increased alkaline phosphatase activity in HT-29 cells and enhanced sucrase activity in differentiated Caco-2 cells, suggesting that specific HMOs contribute to gastrointestinal maturation and digestive function via activation of brush–border enzymes [13]. Supporting these findings, Azagra-Boronat et al. reported that 2′-FL administration in rats increased villus height and surface area after 8 days of intake, indicating an expanded absorptive surface and a potential structural basis for improved nutrient uptake [68]. Together, these studies suggest that 2′-FL may modulate host factors such as mucus layer composition, brush–border enzyme activity, and villus morphology, thereby influencing the efficiency of nutrient absorption. However, as noted by Paone et al. [67], inhibitory effects, including reduced expression of proteins involved in protein digestion and absorption, have also been reported, indicating that the effects of 2′-FL are not uniformly promotive. Taken together, these observations should therefore be interpreted with caution and cannot yet be regarded as providing a sufficient basis for firm nutritional recommendations regarding HMO supplementation in humans. Current evidence for direct HMO–host interactions remains limited to cellular and animal models, and causal relationships in humans have yet to be established. Future research should prioritize well-designed clinical trials to determine the specific effects of HMOs on nutrient absorption in humans.

## 5. Large Intestine Section

### 5.1. Prebiotic Effects of HMOs

As HMOs reach the large intestine, their journey through the gastrointestinal tract concludes, and their prebiotic activity becomes most prominent. In the adult colon, HMOs act as selective substrates for specific gut microbes. Thongaram et al. showed, using an in vitro culture system with HMOs as the sole carbon source, that *Bifidobacterium longum* subsp. *infantis* and *B. breve* efficiently utilized 2′-FL and LNnT, whereas lactobacilli showed limited utilization [8]. Sanchez-Gallardo et al. further demonstrated that members of the *B. pseudocatenulatum* group metabolize various HMOs, particularly LNT, with fucosylated HMO utilization varying by strain. Genomic analysis identified distinct glycosyl hydrolase families linked to this capacity [69]. Onodera et al. demonstrated that HMOs enhance butyrate production through cross-feeding in a co-culture system of *B. bifidum* and *Faecalibacterium prausnitzii*. Specifically, 2′-FL and 3′-SL increased butyrate production efficiency, while 6′-SL promoted the growth of *F. prausnitzii* [70]. These findings indicate that HMOs influence microbial interactions and metabolic networks rather than merely serving as substrates. Recent studies have increasingly employed advanced in vitro systems that simulate the human gastrointestinal tract. Among these, the Simulator of the Human Intestinal Microbial Ecosystem (SHIME) system mimics human physiological conditions from the stomach to the colon and maintains a stable gut microbiome over extended periods, enabling dynamic analysis of microbial composition and metabolism within complex bacterial consortia [71]. Using the SHIME system, Sato et al. established long-term cultures of adult fecal microbiota and showed that co-incubation with HMOs promoted the growth of specific beneficial bacteria and increased propionate and butyrate concentrations [72,73]. These findings suggest that HMOs modulate SCFA-producing bacterial groups within complex gut microbial communities. In terms of SCFAs, Koh et al. [74] summarized that SCFAs such as acetate, propionate, and butyrate—produced by gut microbial fermentation of dietary fiber—exert multiple health benefits. SCFAs serve as an energy source for colonocytes, strengthen the intestinal barrier, and modulate immune tolerance and anti-inflammatory responses. Acting through receptors such as GPR41/43 and GPR109A, they also regulate lipid metabolism, insulin sensitivity, and energy balance, while butyrate acts as a histone deacetylase inhibitor with anti-inflammatory and antitumor effects [74]. Through SHIME experiments, Šuligoj et al. showed that 2′-FL and LNnT supplementation increased *Bifidobacterium* abundance and elevated acetate and butyrate levels, demonstrating that fucosylated HMOs retain bifidogenic selectivity even in the adult gut [75]. Similarly, Pham et al. found that under FODMAP-restricted conditions, combined supplementation with 2′-FL and LNnT restored SCFA levels and increased *Lachnospiraceae* abundance, highlighting the SHIME model’s utility for analyzing microbial responses under specific nutritional conditions [76]. Clinical studies have also supported these observations. Elison et al. administered up to 20 g/day of 2′-FL and LNnT for 2 weeks to 100 healthy adults, confirming safety and tolerability while observing increased *Bifidobacterium* abundance and a reduction in *Firmicutes* [77]. Iribarren et al. reported that 10 g/day of 2′-FL and LNnT (4:1 ratio) for 4 weeks in patients with irritable bowel syndrome (IBS) increased *Bifidobacterium* abundance in both fecal and mucosal samples and altered the metabolic profile [78]. Jacobs et al. found that a 7-day administration of an HMO mixture (up to 18 g/day) in 32 healthy adults induced a dose-dependent increase in *Bifidobacterium*, followed by a shift toward *Bacteroides*, suggesting that combined HMOs induce broader compositional changes than individual compounds [79]. Collectively, these findings indicate that HMOs act as prebiotics by selectively promoting primary degraders such as *Bifidobacterium*, which produce acetate as a key metabolite supporting cross-feeding and enabling butyrate and propionate production by taxa such as *Lachnospiraceae*. Because each oligosaccharide favors distinct microbial taxa and metabolic pathways, HMOs represent a class of prebiotics capable of strategically modulating the adult gut microbiome.

### 5.2. Gut-Modulatory Effects of HMOs

HMOs have been shown to exert gut-modulatory effects in various models of colitis. In a dextran sodium sulfate (DSS)-induced colitis model, Yao et al. demonstrated that 2′-FL ameliorated body weight loss and inflammation scores while protecting the intestinal barrier by restoring *Muc2* and *Nlrp6* expression and suppressing TLR4-related inflammatory pathways [80]. Similarly, Kim et al. reported that 2′-FL and 3-FL restored the expression of *Zo-1* and *Occludin* and reduced IL-6 and TNF-α levels in DSS-induced colitis, supporting their dual roles in inflammation suppression and barrier repair [81]. Grabinger et al. reported that oral supplementation with 2′-FL in an IL-10-deficient Crohn’s disease-like mouse model alleviated inflammation scores and diarrhea symptoms, along with alterations in the gut microbiota characterized by an increase in *Ruminococcus gnavus* and elevated SCFA levels [82]. In neonatal NEC models, Sodhi et al. showed that 2′-FL inhibited TLR4 signaling, reduced histological injury and apoptosis, and mitigated NEC-associated neurological impairment [57,83]. In contrast, Fuhrer et al. found that mice fed milk containing 3′-SL during the suckling period showed enhanced protection against DSS-induced colitis in adulthood. This finding suggests that sialylated HMOs such as 3′-SL may exert long-term protective effects against intestinal inflammation, possibly through epigenetic mechanisms that influence intestinal and immune development during early life [84]. Collectively, these findings emphasize that fucosylated HMOs should be regarded as key components, whereas sialylated HMOs should be selectively utilized depending on the specific disease context. Clinical evidence also supports the gut-modulatory potential of HMOs. Palsson et al. conducted a multicenter open-label trial across 17 U.S. sites involving 317 patients with IBS who received a combination of 2′-FL and LNnT (4:1 mix, 5 g/day) for 12 weeks. Intake of this HMO mixture significantly improved IBS Symptom Severity Scale and Quality-of-Life scores, with symptom relief evident by week 4 and consistent across all IBS subtypes. Improvements were observed in stool consistency, abdominal pain, and bloating [85]. Ryan et al. examined the gut-modulatory effects of 2′-FL both in vitro and in clinical studies involving adults with IBS and ulcerative colitis [86]. In vitro supplementation increased *Bifidobacterium* and butyrate-producing bacteria, elevating acetate and butyrate levels. In the clinical trial, a 2′-FL-containing nutritional formula (4 g/day for 6 weeks) improved Gastrointestinal Quality of Life Index scores and increased fecal abundance of *Bifidobacterium* and *F. prausnitzii*, along with higher fecal butyrate concentrations, indicating concurrent improvements in gut environment and patient well-being. Jackson et al. conducted a randomized controlled trial (EFFICAD) with 92 healthy adults receiving either 2′-FL alone or in combination with inulin for 4 weeks. Stool consistency was assessed using the Bristol Stool Form Scale and bowel movement frequency showed a trend toward normalization in the 2′-FL group [87]. Although the efficacy and safety of 2′-FL have been demonstrated in patients with IBS and healthy adults, clinical data on the gut-modulatory effects of sialylated HMOs remain limited. Further studies are required to determine the therapeutic potential of diverse HMO structures, particularly sialylated species, in human gastrointestinal disorders.

### 5.3. Colonic Barrier-Enhancing Effects of HMOs

Among HMOs, 2′-FL has been shown to strengthen the colonic barrier through multiple mechanisms involving both the mucus layer and tight junctions. Yao et al. reported that in the human colonic goblet cell line LS174T under inflammatory conditions, 2′-FL restored mucin secretory function by inducing MUC2-related genes via NLRP6 signaling [88]. Natividad et al. evaluated HMO-mediated barrier protection using a fermentation model based on infant fecal microbiota (baby M-SHIME^®^) combined with a co-culture system of human colonic epithelial cells [89]. Fermentation products of HMOs significantly attenuated reductions in transepithelial electrical resistance and the increase in FITC-dextran permeability under inflammatory stimulation, thereby preserving barrier integrity. HMO mixtures, compared with individual HMOs or lactose, stimulated the growth of bifidobacteria and butyrate-producing bacteria and produced stronger barrier-enhancing effects. Šuligoj et al. further investigated HMOs using the SHIME model inoculated with adult gut microbiota combined with a human colonic organoid-derived Colon Intestine-Chip [75]. Fermentation products derived from 2′-FL reduced FITC-dextran permeability and, after prolonged fermentation, enhanced barrier function, as reflected by increased *Cldn5* expression. Suppression of the pro-inflammatory cytokine IL-6 was also observed, suggesting that HMOs reinforce the adult colonic barrier through coordinated structural and anti-inflammatory mechanisms. Collectively, in vitro findings support the concept that HMOs act directly to enhance colonic barrier integrity. Animal studies provide complementary evidence for these protective effects. Azagra-Boronat et al. reported that 2′-FL supplementation in a rat model of rotavirus infection reduced markers of barrier disruption, increased *Cldn2* expression, and normalized *Muc2* levels, mitigating rotavirus-induced barrier breakdown [90]. In a DSS-induced colitis mouse model, Dong et al. showed that 2′-FL administration significantly attenuated epithelial damage and colonic permeability, normalized tight junction protein expression, and reduced the oxidative stress marker CYP2E1. Markers of barrier breakdown, including HAPTOGLOBIN and SERPINA3N, were also reduced, indicating barrier repair and reduced inflammation [91]. These molecular and microbial changes suggest that 2′-FL reinforces the colonic barrier at multiple levels by modulating the gut environment and local immune responses. To date, no human clinical trials have directly evaluated the effects of HMOs on large intestinal barrier function as a primary outcome. Existing intervention studies have primarily focused on functional endpoints, such as gut microbiota composition, stool characteristics, gastrointestinal symptoms, and quality of life without directly assessing barrier integrity. Furthermore, clinical data on intestinal modulation by HMOs in humans are largely limited to fucosylated structures such as 2′-FL, whereas evidence for sialylated HMOs remains scarce. These potentially distinct effects are likely driven, at least in part, by the fact that different groups of gut bacteria preferentially utilize each HMO structure and by the divergent functional roles of their constituent monosaccharides, fucose and sialic acid. Future clinical trials should incorporate currently available, and further validated, biomarkers such as zonulin levels and tight junction protein expression as predefined primary outcomes, to clarify the efficacy of different HMO structures in maintaining and restoring colonic barrier function.

## 6. Conclusions

HMOs are resistant to human digestion and traverse the gastrointestinal tract largely intact, exerting beneficial effects at each segment (Figure 2). Their diverse functions underscore both their scientific significance and growing potential. In the oral cavity, HMOs inhibit adhesion and biofilm formation by cariogenic bacteria and *Candida*, supporting oral microbiota homeostasis. In the stomach, sialylated HMOs primarily interfere with *H. pylori* adhesion to the mucosa, indicating their potential as adjuncts for infection prevention. Within the small intestine, HMOs strengthen barrier function by enhancing brush–border enzyme activity, promoting mucin secretion, and maintaining tight junction integrity. These actions collectively contribute to the regulation of inflammation, infection, and allergic responses. In the large intestine, HMOs selectively promote the growth of bifidobacteria and butyrate-producing bacteria, improving the gut environment and reinforcing barrier integrity through SCFA production and cross-feeding interactions. Future research should refine the understanding of species-specific differences in target microbes, barrier molecules, and immune responses and elucidate mechanistic connections across gastrointestinal regions, from the oral cavity to the large intestine. Well-designed clinical trials are urgently needed to explicitly define intestinal barrier function as a primary endpoint and to employ validated and emerging markers such as zonulin, the lactulose/mannitol test, fecal MUC2, and tight junction protein expression as key tools for quantifying intervention effects in humans. HMOs, unique to human milk, combine the positive perception of being human-derived with an established safety profile. Advances in synthesis technology have enabled large-scale production of key HMOs, including 2′-FL, 3′-SL, and 6′-SL, expanding their application beyond infant nutrition to functional foods and medical nutrition for adults. With growing recognition of the gut–organ axis, it is evident that gut composition and microbial activity are closely linked to human health and well-being. Accordingly, HMOs are expected to play an expanding role in supporting gastrointestinal integrity and function across all life stages, addressing both clinical and societal health challenges.

## Figures and Tables

**Figure 1 microorganisms-14-00029-f001:**
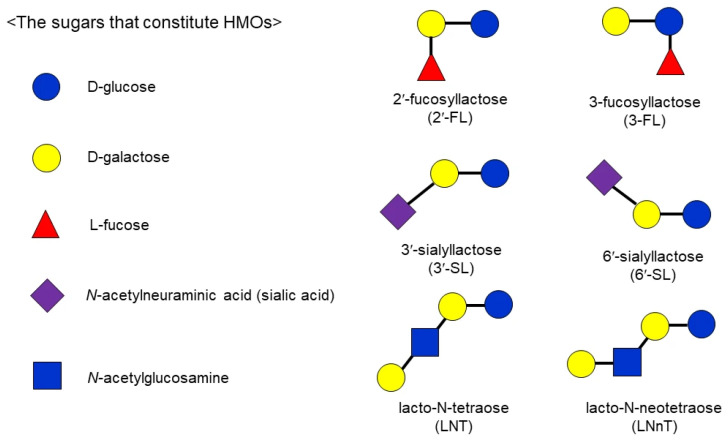
Components of major human milk oligosaccharides.

**Figure 2 microorganisms-14-00029-f002:**
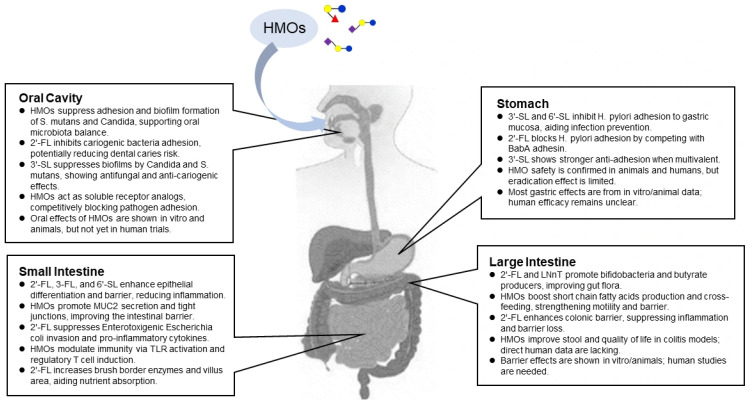
Physiological effects of HMOs along the gastrointestinal tract.

## Data Availability

No new data were created or analyzed in this study. Data sharing is not applicable to this article.

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
