# Peer review of "Gastrointestinal Journey of Human Milk Oligosaccharides: From Breastfeeding Origins to Functional Roles in Adults"

_microorganisms, 2025, doi:10.3390/microorganisms14010029_

Round 1
Reviewer 1 Report
Comments and Suggestions for Authors
The manuscript entitled “Gastrointestinal Journey of Human Milk Oligosaccharides: From Breastfeeding Origins to Functional Roles in Adults” examined an important topic within the fields of nutrition and the microbiome. Overall, the paper is well-structured and provides valuable insights into the potential dietary role of complex sugar intake in shaping the microbiome and influencing adult health. The topic is scientifically relevant, and the review offers a solid overview that may support the design of future studies investigating the potential applications of HMOs. In general, the manuscript presents a well-founded review that could be of interest to a broad readership.
I have a few minor comments and suggestions that should be addressed before the manuscript can be considered for publication:
The abstract is well-written, clear, and informative.
The introduction provides essential background on HMO structure and its use in infant formulas. However, appropriate reference regarding FDA approval of HMOs as novel food ingredients should be added.
Figures 1 and 2 are informative and appropriately support the discussed content.
References (e.g., 35, 48, 69, 81, and others) should be carefully revised and adjusted according to the journal’s instructions for authors.

Author Response
Comment 1: The manuscript entitled “Gastrointestinal Journey of Human Milk Oligosaccharides: From Breastfeeding Origins to Functional Roles in Adults” examined an important topic within the fields of nutrition and the microbiome. Overall, the paper is well-structured and provides valuable insights into the potential dietary role of complex sugar intake in shaping the microbiome and influencing adult health. The topic is scientifically relevant, and the review offers a solid overview that may support the design of future studies investigating the potential applications of HMOs. In general, the manuscript presents a well-founded review that could be of interest to a broad readership. I have a few minor comments and suggestions that should be addressed before the manuscript can be considered for publication:
Response 1: We sincerely thank the reviewer for their thorough evaluation and positive assessment of our manuscript. We greatly appreciate the recognition of the scientific relevance of this topic and the comments highlighting the structure, clarity, and potential contribution of our review to future research on HMOs and adult health. We have revised the manuscript in response to your comments as outlined below.
Comment 2: The abstract is well-written, clear, and informative.
Response 2: We thank the reviewer for this positive comment on the abstract and are pleased that it was found to be clear and informative. No revisions were made to this section.
Comment 3: The introduction provides essential background on HMO structure and its use in infant formulas. However, appropriate reference regarding FDA approval of HMOs as novel food ingredients should be added.
Response 3: We thank the reviewer for this helpful suggestion. In the Introduction, we have added a sentence describing the regulatory framework for HMOs as food ingredients and explicitly mentioning their approval as novel food ingredients/GRAS substances by regulatory authorities, including the FDA and EFSA. To support this statement on regulatory approval, we have added the following sentence to the Introduction (lines 72–74) together with a new reference: “The regulatory framework governing the use of these HMOs as food ingredients, including safety assessment and conditions of use, has been comprehensively reviewed by Salminen [25].”
Comment 4: Figures 1 and 2 are informative and appropriately support the discussed content.
Response 4: We thank the reviewer for this positive comment on the Figures. No revisions were made to our Figures.
Comment 5: References (e.g., 35, 48, 69, 81, and others) should be carefully revised and adjusted according to the journal’s instructions for authors.
Response 5: We thank the reviewer for this helpful comment. We have carefully revised the entire reference list and adjusted the formatting to conform to the Microorganisms (MDPI) reference style. We corrected the formatting of references 35, 48, 69, 81 and other entries where the citation format previously followed a non-MDPI style.
Reviewer 2 Report
Comments and Suggestions for Authors
The reviewer apologizes for the late review. I read the manuscript several times and tried to find any significant issues which should be resolved before recommending the article for publication. But no such issues were found, therefore, I recommend the review for publication in its current form.
Author Response
Comment 1: The reviewer apologizes for the late review. I read the manuscript several times and tried to find any significant issues which should be resolved before recommending the article for publication. But no such issues were found, therefore, I recommend the review for publication in its current form.
Response 1: We sincerely thank the reviewer for their careful and repeated reading of our manuscript and for the very positive evaluation. We greatly appreciate the recommendation for publication in its current form. In revising the manuscript, we have only made minor adjustments in response to specific comments from the other reviewers (e.g., reference formatting and clarification of certain sections), without altering the overall content or conclusions that were positively assessed in your review.
Reviewer 3 Report
Comments and Suggestions for Authors
The following points should be addressed in the revision:
- Section 2.1 – It must be unequivocally emphasised that the existing conclusions regarding anti-adhesion and anti-caries activity are primarily mechanistic in nature, resulting from in vitro and model studies. In contrast, their actual clinical impact on oral health remains hypothetical due to the lack of large-scale human trials. In the discussion or conclusion, it is important to clearly highlight the existing disparity between the advanced understanding of molecular mechanisms and the limited availability of translational data derived from clinical research.
- Section 3.1 – It is necessary to specify the possible causes of the limited clinical efficacy against H. pylori and to emphasise the need to define dose-effect profiles strongly. In the sections concerning H. pylori and selected prebiotic effects, it is recommended to highlight the need for future research to establish dose-response relationships and investigate the nature of strain-specific impacts, while also drawing attention to the potential influence of individual variability in the human microbiome on the obtained results.
- Sections 4.1 and 4.2 – It is essential to emphasise that future clinical trials should incorporate validated biomarkers as primary endpoints to confirm the gut barrier-enhancing effects observed in preclinical models. There is an urgent need to develop reliable, non-invasive markers to monitor gut barrier function and local immune status in clinical trials, as this currently represents a significant technical limitation.
- Section 4.3 – The conclusions should be formulated cautiously, emphasising that current observations are not unequivocally promotional, and a causal relationship in the human population has not yet been demonstrated.
- Sections 5.2 and 5.3 – A significant lack of clinical data regarding intestinal modulation by sialylated HMOs in human gastrointestinal disorders must be explicitly pointed out, especially when compared to the more readily available data for 2′-FL. It is essential to note that the available clinical data pertain to functional endpoints related to barrier function, rather than direct molecular assessment. It is also recommended to clarify the distinction between the postulated roles of fucosylated and sialylated HMOs, suggesting their selection should be dependent on the clinical context, in line with preclinical findings.
- Section 6 – the need for future clinical trials where gut barrier function serves as the primary endpoint should be unequivocally stated, and the necessity of employing validated biomarkers as crucial tools for assessing intervention effects should be emphasised.
Author Response
Comment 1: Section 2.1 – It must be unequivocally emphasised that the existing conclusions regarding anti-adhesion and anti-caries activity are primarily mechanistic in nature, resulting from in vitro and model studies. In contrast, their actual clinical impact on oral health remains hypothetical due to the lack of large-scale human trials. In the discussion or conclusion, it is important to clearly highlight the existing disparity between the advanced understanding of molecular mechanisms and the limited availability of translational data derived from clinical research.
Response 1: We thank the reviewer for this important comment. In response, we have revised Section 2.1 to emphasize the primarily mechanistic nature of the current evidence more clearly. Specifically, we added the following sentence in lines 123–125: “Thus, at present, conclusions regarding the effects of HMOs on oral health are largely mechanistic and based on findings from in vitro and other experimental model studies, and their actual clinical impact on oral health remains hypothetical.” This addition explicitly highlights the gap between the detailed understanding of molecular mechanisms and the limited availability of translational data from large-scale human trials, as recommended.
Comment 2: Section 3.1 – It is necessary to specify the possible causes of the limited clinical efficacy against H. pylori and to emphasise the need to define dose-effect profiles strongly. In the sections concerning H. pylori and selected prebiotic effects, it is recommended to highlight the need for future research to establish dose-response relationships and investigate the nature of strain-specific impacts, while also drawing attention to the potential influence of individual variability in the human microbiome on the obtained results.
Response 2: We thank the reviewer for this insightful comment. In response, we have revised Section 3.1 to specify potential reasons for the limited clinical efficacy of HMOs against H. pylori and to emphasize the need to better define dose–effect profiles. Specifically, we added the following paragraph in lines 162–173: “The limited clinical efficacy of HMOs observed so far against H. pylori infection is probably influenced by several factors. First, after oral administration, HMOs may not reach or remain at sufficiently high concentrations on the gastric mucosal surface, which could limit their ability to inhibit H. pylori adhesion through competitive binding to host receptors. Second, strain-to-strain differences in H. pylori adhesins and virulence factors, as well as in commensal microbes, are likely to result in variable sensitivity to specific HMOs. Third, when HMOs are administered as adjuncts to standard eradication regimens containing proton pump inhibitors and antibiotics, profound changes in the gastric environment and microbiota may mask any additional HMO-mediated benefits. Therefore, future clinical trials should be designed to rigorously define dose–response relationships for individual HMOs, to characterize strain-specific effects, and to account for inter-individual variability in the human microbiome when interpreting treatment outcomes.” This revision directly addresses the reviewer’s suggestion by outlining plausible causes of the limited clinical efficacy observed to date and by clearly highlighting the need for future studies to establish dose–response relationships, investigate strain-specific effects, and consider inter-individual microbiome variability.
Comment 3: Sections 4.1 and 4.2 – It is essential to emphasise that future clinical trials should incorporate validated biomarkers as primary endpoints to confirm the gut barrier-enhancing effects observed in preclinical models. There is an urgent need to develop reliable, non-invasive markers to monitor gut barrier function and local immune status in clinical trials, as this currently represents a significant technical limitation.
Response 3: We thank the reviewer for this important and constructive comment. In response, we have revised Sections 4.1 and 4.2 to emphasize the need more clearly for validated biomarkers and the current technical limitations in assessing gut barrier function in clinical studies. Specifically, we added the following sentences in lines 196–200: “However, validated, non-invasive biomarkers for monitoring gut barrier integrity in clinical settings remain limited. Clinical trials that incorporate these parameters as predefined endpoints, and that further develop and validate additional biomarkers, are required to clarify the barrier-enhancing effects of HMOs in humans.” These additions explicitly address the reviewer’s recommendation by underlining both the necessity of using validated biomarkers as primary endpoints and the urgent need to develop reliable, non-invasive markers for gut barrier function and local immune status in future clinical trials.
Comment 4: Section 4.3 – The conclusions should be formulated cautiously, emphasising that current observations are not unequivocally promotional, and a causal relationship in the human population has not yet been demonstrated.
Response 4: We thank the reviewer for this important comment and agree that the conclusions in Section 4.3 should be presented with appropriate caution. In response, we have revised the text to more clearly reflect the preliminary nature of the current evidence and the lack of demonstrated causal relationships in humans. Specifically, we added the following sentence in lines 266–268: “Taken together, these observations should therefore be interpreted with caution and cannot yet be regarded as providing a sufficient basis for firm nutritional recommendations regarding HMO supplementation in humans.” This revision explicitly acknowledges that the available findings are not unequivocally promotional, and that current data do not yet support definitive causal conclusions in human populations, in line with the reviewer’s recommendation.
Comment 5: Sections 5.2 and 5.3 – A significant lack of clinical data regarding intestinal modulation by sialylated HMOs in human gastrointestinal disorders must be explicitly pointed out, especially when compared to the more readily available data for 2′-FL. It is essential to note that the available clinical data pertain to functional endpoints related to barrier function, rather than direct molecular assessment. It is also recommended to clarify the distinction between the postulated roles of fucosylated and sialylated HMOs, suggesting their selection should be dependent on the clinical context, in line with preclinical findings.
Response 5: We thank the reviewer for this important and insightful comment. As suggested, we agree that the marked lack of clinical data on intestinal modulation by sialylated HMOs in human gastrointestinal disorders, particularly in comparison with the more abundant evidence for 2′-FL and other fucosylated HMOs, should be explicitly highlighted. We also concur that it is important to point out that the currently available clinical data are mainly based on functional endpoints related to barrier function rather than on direct molecular assessments. In addition, following the reviewer’s recommendation, we have clarified the distinction between the postulated roles of fucosylated and sialylated HMOs, emphasizing that their selection should be tailored to the clinical context in line with preclinical findings. Specifically, we added the following text in lines 401–409: “Furthermore, clinical data on intestinal modulation by HMOs in humans are largely limited to fucosylated structures such as 2′-FL, whereas evidence for sialylated HMOs remains scarce. These potentially distinct effects are likely driven, at least in part, by the fact that different groups of gut bacteria preferentially utilize each HMO structure and by the divergent functional roles of their constituent monosaccharides, fucose and sialic acid. Future clinical trials should incorporate currently available, and further validated, biomarkers such as zonulin levels and tight junction protein expression as predefined primary outcomes, to clarify the efficacy of different HMO structures in maintaining and restoring colonic barrier function.” We believe these revisions directly address the reviewer’s concerns by explicitly acknowledging the imbalance in clinical evidence between fucosylated and sialylated HMOs, the functional (rather than molecular) nature of current endpoints, and the need to consider structure-specific roles of HMOs in a context-dependent clinical framework.
Comment 6: Section 6 – the need for future clinical trials where gut barrier function serves as the primary endpoint should be unequivocally stated, and the necessity of employing validated biomarkers as crucial tools for assessing intervention effects should be emphasised.
Response 6: We thank the reviewer for this important comment and fully agree that the need for future clinical trials with gut barrier function as the primary endpoint, and the central role of validated biomarkers, should be clearly stated. In response, we revised the corresponding part of Section 6 (lines 426–433) rather than simply adding a new sentence. The previous text has been replaced with the following: “Future research should refine the understanding of species-specific differences in target microbes, barrier molecules, and immune responses and elucidate mechanistic connections across gastrointestinal regions, from the oral cavity to the large intestine. Well-designed clinical trials are urgently needed to explicitly define intestinal barrier function as a primary endpoint and to employ validated and emerging markers such as zonulin, the lactulose/mannitol test, fecal MUC2, and tight junction protein expression as key tools for quantifying intervention effects in humans.” This revision explicitly states the need to prioritize gut barrier function as a primary endpoint in future clinical trials and highlights the necessity of using validated biomarkers as crucial tools for assessing intervention effects, in line with the reviewer’s recommendation.
Round 2
Reviewer 3 Report
Comments and Suggestions for Authors
I believe the manuscript has been sufficiently revised to warrant publication. The authors have addressed all six reviewer comments in a detailed and constructive manner, implementing key modifications that temper the conclusions and explicitly highlight the gaps between mechanistic evidence and clinical data. The changes are precise, fully address all critical concerns, and significantly enhance the scientific and clinical rigour of the manuscript.